# Research on Ecological Effect Assessment Method of Ecological Restoration of Open-Pit Coal Mines in Alpine Regions

**DOI:** 10.3390/ijerph19137682

**Published:** 2022-06-23

**Authors:** Meng Yuan, Jingyi Ouyang, Shuanning Zheng, Ye Tian, Ran Sun, Rui Bao, Tao Li, Tianshu Yu, Shuang Li, Di Wu, Yongjie Liu, Changyou Xu, Yu Zhu

**Affiliations:** 1Life Science College, Fujian Agriculture and Forestry University, Fuzhou 350002, China; myuan@iue.ac.cn; 2Institute of Urban Environment, Chinese Academy of Sciences, Xiamen 361021, China; jyouyang@iue.ac.cn (J.O.); snzheng@iue.ac.cn (S.Z.); 3State Key Laboratory of Urban and Regional Ecology, Research Center for Eco-Environmental Sciences, Chinese Academy of Sciences, Beijing 100085, China; ransun@rcees.ac.cn (R.S.); ruibao_st@rcees.ac.cn (R.B.); taoli_st@rcees.ac.cn (T.L.); tsyu@rcees.ac.cn (T.Y.); shuangliu_st@rcees.ac.cn (S.L.); 4Institute of Architecture Design and Research, Chinese Academy of Sciences, Beijing 100086, China; zhongw@adcas.cn; 5SPIC Nei Mongol Energy Co., Ltd., Tongliao 028011, China; liuyongjie@spic.com.cn (Y.L.); xuchangyou@spic.com.cn (C.X.); 6China IPPR International Engineering Co., Ltd., Beijing 100089, China; zhuyu@ippr.net

**Keywords:** open-pit coal mines in alpine regions, ecological restoration, ecological effects assessment method, Baiyinhua area

## Abstract

Open-pit mining is an important form of coal mining in China, and its damage to the ecological environment is particularly obvious in alpine regions. The ecological restoration of alpine open-pit coal mines faces severe challenges, and its restoration effect will directly affect the ecological security of China. Meanwhile, comprehensive and system-oriented evaluation of ecological restoration effects is still insufficient in current research. In this study, we selected different quantities of assessment factors on the two scales of ecological project area and ecological impact area to evaluate the ecological restoration effect of an alpine open-pit coal mine. Then, we formed a multi-scale and multi-dimensional ecological restoration effect assessment model of the alpine open-pit coal mine and used this model to analyze the implementation effect of the ecological restoration project of the Baiyinhua No. 2 Open-pit Mine. The results show that the multi-scale and multi-dimensional ecological restoration effect assessment model of alpine open-pit coal mine proposed in this study can accurately characterize the restoration effect of open-pit coal mines in alpine regions and can also be used as a significant evaluation tool in the future ecological construction of mining areas. This study hopes the multi-scale and multi-dimensional ecological restoration effect assessment model of alpine open-pit coal mine can provide a comprehensive, systematic, and scientific evaluation method for the ecological restoration of alpine open-pit coal mines and provide a scientific basis for the ecological restoration and green development of relevant mining areas.

## 1. Introduction

Coal is the main energy source in China, and its dominant position will not change in the short term [1]. Open-pit mining is an important coal mining method in China. Most of China’s open-pit coal mines are concentrated in the northern arid and semi-arid regions [2]. The background conditions of the ecological environment in these areas are mostly poor, and serious soil erosion there will cause a series of secondary disasters, which will bring many adverse effects on the safety people and property in the mining areas [3]. However, the ecological restoration challenges that the open-pit coal mines in the alpine regions of northern China face are even more severe, and the ecological restoration effects of these mines of these areas will directly affect China’s ecological security [4].

Mine ecological restoration falls under the scope of ecological restoration. China’s mine ecological restoration can be traced back to the 1950s. As an important segment to measure the implementation effect of ecological restoration projects, ecological restoration effect assessment has made some progress in both theory and practice. For the assessment of the effect of mine ecological restoration, the previous studies have mainly focused on mining wasteland [5], tailings pond [6], pit slope [7], etc., and the assessment methods mainly included remote sensing monitoring, soil evaluation, model evaluation, etc. For example, Huang et al., used the combination of remote sensing and GIS to monitor and assess the vegetation restoration of coal mine [8]; the topoedaphic unit analysis proposed by Krabbenhoft et al., compared the soil factors and vegetation factors in the restoration area and the reference area in order to assess the ecological restoration quality [9]; and Ma et al., constructed the ecological assessment model of a large open-pit mining area based on the theory and method of landscape ecological assessment [10].

However, most of the assessment methods in these studies focus on a single indicator, such as fractional vegetation cover, soil, or landscape. Similarly, the assessment areas of previous research were also very limited, mostly focusing on the assessment of the restoration area or mining area and not raising the vision of evaluation to a more macro-scale perspective. Therefore, the comprehensive and system-oriented assessment of ecological restoration effects is still insufficient [11]. Additionally, most of the current mine ecological restoration efforts are concentrated in low- and medium-altitude areas, and there is insufficient attention to alpine areas, so the effect assessment of ecological restoration projects is very limited in this area [12].

In order to make up the gap in existing research, this study formed a multi-scale and multi-dimensional ecological restoration effect assessment model of an alpine open-pit coal mine and used this model to analyze the implementation effect of the ecological restoration project of the Baiyinhua No. 2 Open-pit Mine. This model can better reflect the impact of ecological restoration projects. At the same time, this study also aims to enrich the research on the assessment of the effect of ecological restoration of open-pit coal mines in alpine areas. Specifically, two parts were completed: (1) we selected four indicators, i.e., fractional vegetation cover, net primary productivity, soil erosion, and carbon sequestration effect; and (2) we divided two different dimensions of scales in the study area, namely the ecological project area and the ecological impact area. The scope is on a more macro level, and the impact of the ecological restoration project on the environs was fully considered, so as to form the multi-scale and multi-dimensional ecological restoration effect assessment model of an alpine open-pit coal mine and to evaluate the implementation effect of the ecological restoration project in the study area. Consequently, this study can provide a reference for the research related to the comprehensive assessment of ecological restoration in alpine regions and mines.

## 2. Materials and Methods

### 2.1. Study Area

The common conditions in alpine regions, such as low temperature, high altitude, windy and dusty weather, serious soil erosion, and special planting situations, have restricted the ecological restoration of open-pit coal mines in the region. The Baiyinhua No. 2 Open-pit Mine originated from the Baiyinhua Coalfield. It is located on the western slope of the southern section of the Greater Khingan Mountains in central and eastern Inner Mongolia, in the hinterland of the Xilingol Grassland, and is affiliated to the Xilingol League West Ujimqin Banner [13]. The coal field is distributed from the northeast to the southwest, covering an area of 510 km^2^. Of this, the open-pit stope area is 6.28 km^2^, the maximum mining depth is 227 m, and the lowest mining elevation is 854 m; the total area of the dump is 16.02 km^2^. The local annual average temperature is 2.2 °C, the annual average frost-free period is 105 days, the maximum frozen soil depth is 327 cm, the annual average precipitation is 340.4 mm, and the annual average evaporation is 1760.8 mm. The continental climate there is obvious, with a cold and long winter and spring, making it a typical alpine open-pit coal mine [14,15]. The geographical location and loam conditions of the Baiyinhua No. 2 Open-pit Mine are different from those of other open-pit mines in the industry. For the Baiyinhua No. 2 Open-pit Mine, its massive scale, alpine location, and in urgent needs of ecological restoration but meanwhile lacks sufficient cases as its reference. All these above pose great challenges to its green construction and green development [16]. Since 2018, the Baiyinhua No. 2 Open-pit Mine has successively carried out ecological restoration and governance work with the general requirements of “making a major change in one year and thorough rectification in three years and building a benchmark for green mines in China’s alpine regions” and achieved remarkable results. It fills the current gap in the ecological restoration model of open-pit coal mines in alpine regions. Our study selected this area to assess the evaluation of the effect of the ecological restoration project because it has a strong representative significance to make up the deficiency in the research of the effect assessment of open-pit coal mine ecological restoration project in alpine areas. The location of the open-pit mine is shown in Figure 1.

This study obtained the required data by collecting and consulting the territorial spa-tial planning and statistical yearbook, field investigation, and remote sensing monitoring in the study area. Specifically, three sources were consulted: (1) satellite image data of the Xilingol League (https://earth.google.com/, accessed on 2 September 2021); (2) Landsat 8 remote sensing images of Inner Mongolia Autonomous Region (https://www.gscloud.cn/, accessed on 3 September 2021), with a spatial resolution of 30 m × 30 m; and (3) land use data of the Xilingol League from 2017 to 2021 (http://www.dsac.cn/, accessed on 11 January 2022), with a spatial resolution of 15 m × 15 m. The remote sensing image of Baiyinhua No. 2 Open-pit Mine is shown in Figure 2.

### 2.2. Research Methods

In view of the current situation in which the evaluation scale or evaluation dimension is relatively single in the evaluation of ecological restoration projects and it is difficult to evaluate the comprehensive effect of the restoration, this study proposed a multi-scale and multi-dimensional ecological restoration effect assessment model of an alpine open-pit coal mine. The model, formed from the perspective of ecological security, ecological health, and ecological well-being, includes two different scales and four different indicators. For different scales, we used different indicators to evaluate the effects of ecological restoration, and the evaluation indicators corresponding to each scale were identified as a dimension in order to get direct and intuitive results. The most important role of the model is to test whether the ecological restoration of the alpine open-pit coal mine has achieved the effect of maintaining, improving, and enhancing the service capacity of the regional ecosystem.

The two scales in this model are ecological project area and ecological impact area. The total area of the Baiyinhua No. 2 Open-pit Mine dump is 16.02 km^2^, of which, the total ecological project area is 6.85 km^2^. According to the actual situation of the restoration project and the interpretation of remote sensing, the area in the dump with a vegetation coverage of less than 5% is now defined as non-ecological project area, and the rest of the area is the ecological project area. The ecological impact area refers to an area that is greatly affected by coal mining and ecological restoration of its dumps. In this study, the three-kilometer area around Baiyinhua No. 2 Open-pit Mine (including the dumps) is defined as the ecological impact area.

The four indicators that make up the dimensions are fractional vegetation cover, net primary productivity, soil erosion, and carbon sequestration effect. Among them, fractional vegetation cover and water and soil erosion indicators were selected from the perspective of ecological security; net primary productivity was selected from the perspective of ecological health; and carbon sequestration effect was selected from the perspective of ecological well-being. As land reclamation and virescence are the most intuitive and important effects of ecological restoration, the indicators included in the assessment dimension corresponding to the ecological project area were fractional vegetation cover and carbon sequestration effect. In the assessment dimension corresponding to the ecological impact area, we added net primary productivity and soil erosion as two indicators on the basis of fractional vegetation cover and carbon sequestration effect, to more comprehensively evaluate the effect of ecological restoration. The ecological restoration effect assessment model of an alpine open-pit coal mine is shown in Figure 3.

#### 2.2.1. Fractional Vegetation Cover

Vegetation restoration is an important part of ecological restoration in mining area [17]. Fractional vegetation cover (FVC) is an important parameter to describe the ecosystem and reflect the distribution characteristics of surface vegetation; it is also an important indicator to reflect the ecological restoration of mines [18]. The method of remote sensing estimation of vegetation coverage in the study area in this study mainly refers to the model, which after improving the parameter estimation of the pixel dichotomy, was established by Li et al., to quantitatively estimate fractional vegetation cover using the normalized vegetation index NDVI [19].

The formula for the calculation of fractional vegetation cover based on NDVI is as follows:Fc = (NDVI − NDVIsoil)/(NDVIveg − NDVIsoil),(1)
NDVI = (NIR − R)/(NIR + R)(2)

In the formula, Fc is the vegetation coverage based on the NDVI method; NDVIsoil is the NDVI value of completely bare soil or area with no vegetation coverage; and NDVIveg is the NDVI value of pixels completely covered by vegetation, that is, the NDVI value of pure vegetation pixels. This study calculated the fractional vegetation cover based on the Map Algebra tool of the ArcGIS software platform.

#### 2.2.2. Net Primary Productivity

Net primary productivity (NPP) refers to the net accumulation of organic matter per unit time and unit area of plants, which can reflect the quality of the regional natural ecological environment [20]. The study calculated the regional net primary productivity based on the evaluation results of the regional vegetation index and the light energy utilization model. The following formula was used:y = 3807.2x^3.0711^(3)

In the formula, y represents net primary productivity and x represents vegetation index. This study calculated the net primary productivity based on the Map Algebra tool of ArcGIS software platform and land use data.

#### 2.2.3. Soil Erosion

Soil erosion refers to the whole process of soil erosion, transportation, and sedimentation under the action of water flow [21]. In this study, we used the grid data set of annual rainfall erosivity in China, combine with the universal soil loss equation (USLE) model, to calculate the amount of soil erosion before and after the implementation of the restoration project, and then analyzed the effect of the ecological restoration project on soil erosion [22,23].

The formula used was
A = R × K × LS × C × P(4)

In the formula, A represents annual soil erosion [t/(hm^2^·a)]; R represents rainfall erosivity factor [MJ·mm/(hm^2^·h·a)]; K represents soil erodibility factor [t·h/(MJ·mm·a)]; LS stands for slope length and gradient, dimensionless; C stands for vegetation cover factor, dimensionless; and P stands for soil and water conservation measure factor, dimensionless.

#### 2.2.4. Carbon Sequestration Effect

Carbon sequestration and oxygen release is an important function of ecosystem services [24]. According to the reaction equation of plant photosynthesis and respiration, it can be known that 1.62 g of CO_2_ is required to form 1 g of dry matter, and during this process, 1.2 g of O_2_ is released. Based on this, the carbon sequestration in different regions can be estimated [25,26].

In this study, the InVEST-Carbon model was used to calculate the amount of carbon sequestration in the ecological project area, the ecological impact area, and the administrative region and to evaluate the carbon sequestration effect of the ecosystem after the implementation of the ecological restoration project. The data sources include self-test data and data provided by local governments.

## 3. Results

### 3.1. Ecological Project Area

From 2017 to 2021, the vegetation coverage in the ecological project area of the Baiyinhua No. 2 Open-pit Mine increased from 45.84% to 63.77%, with an increment of 17.93% and a growth rate of 39.11%; carbon sequestration increased from 0.341 t to 0.357 t, and the growth rate was 4.69%. Details are shown in Figure 4 and Figure 5.

### 3.2. Ecological Impact Area

In the ecological impact area, the vegetation coverage increased from 42.00% to 47.89%, with an increment of 5.89% and a growth rate of 14.02%; the increment of net primary productivity per unit area was 6.55 g·C/m^2^ and the growth rate was 2.58%; the soil erosion modulus decreased by 113.38 t/hm^2^·a, with a decrease rate of 2.09%; the amount of carbon sequestration increased from 5.31 t to 5.67 t, an increase of 0.46 t and a growth rate of 8.66%. Details can be seen in Figure 6, Figure 7 and Figure 8.

## 4. Discussion

### 4.1. Intuitiveness and Effectiveness of the Multi-Scale and Multi-Dimensional Ecological Restoration Effect Assessment Model of Alpine Open-Pit Coal Mine for Ecological Restoration Effect Evaluation

In this study, we paid close attention to the fractional vegetation cover, net primary productivity, soil erosion, and carbon sequestration effect of the restoration area from 2017 to 2021, which effectively reflects the effects of the ecological restoration project. Through comparison, it can be found that there are obvious regional differences in fractional vegetation cover and carbon sequestration between different ecological project areas, which are mainly affected by different restoration measures and management modes in each ecological project area. The fractional vegetation cover and carbon sequestration in the north dump ecological project area decreased to a certain extent, because the northeast part of the north dump is the main dumping area at present. The dumping soil there occupies a large area of grassland vegetation, and at the same time, the growth of the surface vegetation in some areas where the restoration was carried out was incomplete. In addition, since vegetation restoration was not carried out in the central part of the north dump and the northwest of the south dump, there is no vegetation coverage or a low degree of vegetation coverage. In the ecological impact area, the northern part was affected by continuous mining and, thus, with more human activity interference, especially in its newly-developed mining area, both fractional vegetation cover and net primary productivity decreased to varying degrees. The situation of soil erosion in the ecological impact area is still severe, but the implementation of the ecological restoration project has played a certain role in alleviating it. In general, the ecosystem service capacity of the restoration area is currently in the situation of maintenance, improvement, and enhancement. In the process of comparison with the same regional research [27], we can find the similarities of the research results, which further confirms the credibility of our research results.

The above aspects can explain that the multi-scale and multi-dimensional ecological restoration effect assessment model of an alpine open-pit coal mine adopted in this study can directly and entirely reflect the implementation effect of the restoration project and can carry out continuous observation and evaluation to illustrate the long-term effect of the restoration project. Meanwhile, this study assesses the implementation effect of ecological restoration project on a more macro dimension of the ecological impact area, which is a breakthrough compared with the previous research, which only focused on the restoration effect within the mining area.

### 4.2. Limitations and Future Research Directions

By selecting two different levels of scales and two dimensions covering four different indicators, this study formed a multi-scale and multi-dimensional ecological restoration effect assessment model of an alpine open-pit coal mine and evaluated the effect of the ecological restoration project of the Baiyinhua No. 2 Open-pit Mine, providing a new model for the effect evaluation of ecological restoration, and enriching the research on the ecological restoration effect evaluation of open-pit coal mines in alpine areas.

However, there are also some shortcomings. For example, the selection of assessment indicators was affected by data availability and richness, resulting in a certain subjectivity in the selection of indicators, which can be improved in this regard.

Under the new situation, the basic concepts, institutional mechanisms, definitions, and missions of mine ecological restoration have been expanded and innovated to varying degrees, which has placed higher requirements on ecological restoration in the traditional sense [28]. Mine ecological restoration cannot be limited to a small scale and a single dimension but needs to be raised to the macro-scale perspective of territorial–spatial–ecological restoration, considering the coupling with other subsystems such as the socio-economic system and the humanistic system, and focusing on the integrity, systematism, and comprehensiveness of restoration [29,30,31]. This will also be the main direction of future improvement of our study.

Based on the existing research, the most critical problem in the study field of mine ecological restoration assessment is still the lack of industrial evaluation standards [26,31]. The lack of universal evaluation standards will impose certain restrictions on the evaluation of the effectiveness of mine ecological restoration and on the supervision of the whole process, which is not conducive to scientific evaluation and dynamic monitoring of the level and effect of mine ecological restoration work across the country [32]. Therefore, in future research and development, it is necessary to actively promote the establishment of an authoritative, scientific, and operable evaluation index system, and on this basis, according to the relevant characteristics of each industry and each region, to gradually set up differentiated evaluation indicators for each industry and region to strengthen their pertinence [33].

In addition, the establishment of dual carbon goals has put forward higher requirements for the ecological restoration of mines and also promoted the further improvement and innovation of ecological restoration assessment work [34,35]. The goal of peaking carbon emissions and achieving carbon neutrality has accelerated the arrival of the post-mining era, and in the post-mining era, one of the most important tasks is to realize the ecological restoration of mines [36]. At this stage, many experts and scholars have analyzed the development prospects and challenges of mine ecological restoration under this background and proposed a new model of mine ecological restoration for carbon neutrality [37]. From the evaluation point of view, in future, relevant indicators of the carbon footprint can be introduced into studies on the assessment of the effect of mine ecological restoration. Taking the carbon sink increment as a quantifiable, assessable, and predictable indicator of mine ecological restoration results can better combine mine ecological restoration work with national policies and strategies and lead to a timely response to actual needs [38].

## 5. Conclusions

This study formed a multi-scale and multi-dimensional ecological restoration effect assessment model of an alpine open-pit coal mine, which better reflects the impact of the ecological restoration project on the region. At the same time, we selected the Baiyinhua No. 2 Open-pit Mine in the West Ujimqin Banner as the main area of the study, which also makes up for the deficiency in the research on the assessment of the effect of ecological restoration of open-pit coal mines in alpine regions.

(1)The multi-scale and multi-dimensional ecological restoration effect assessment model of an alpine open-pit coal mine can intuitively and effectively reflect whether the ecological restoration projects have achieved the effect of maintaining, improving, and enhancing the service capacity of the regional ecosystem and provide a scientific basis for the ecological restoration and green development of open-pit coal mines in alpine regions.(2)According to the interpretation and analysis of remote sensing, the total area of the dump site of the Baiyinhua No. 2 Open-pit Mine is 16.02 km^2^, of which the total area of the ecological restoration project is 6.85 km^2^ and the restoration rate is 55.96%. In the ecological project area, the overall fractional vegetation cover and carbon sequestration showed an increasing trend. The fractional vegetation cover of each project area gradually increased with the passage of the growing season. In the ecological impact area, the overall vegetation coverage, net primary productivity, and carbon sequestration all showed an increasing trend, and the soil erosion modulus also decreased to a certain extent. The ecosystem service capacity of the Baiyinhua No. 2 Open-pit Mine is in the situation of maintenance, improvement, and enhancement. The above results further prove the effectiveness of the Baiyinhua No. 2 Open-pit Mine ecological restoration.

## Figures and Tables

**Figure 1 ijerph-19-07682-f001:**
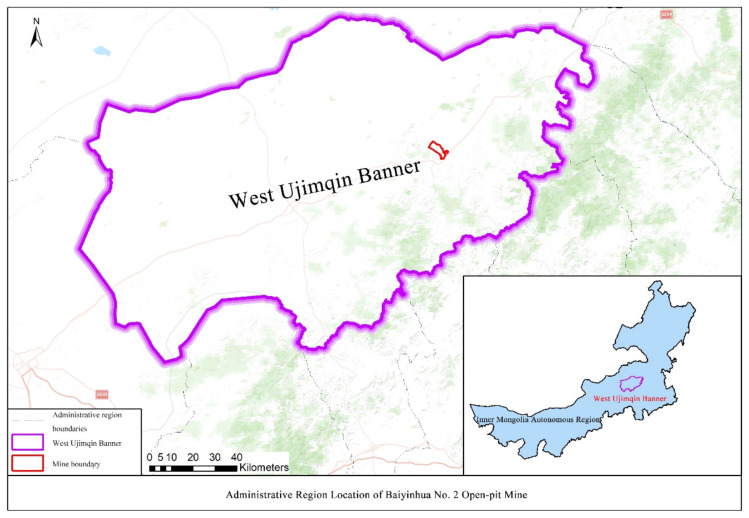
Administrative region location of Baiyinhua No. 2 Open-pit Mine.

**Figure 2 ijerph-19-07682-f002:**
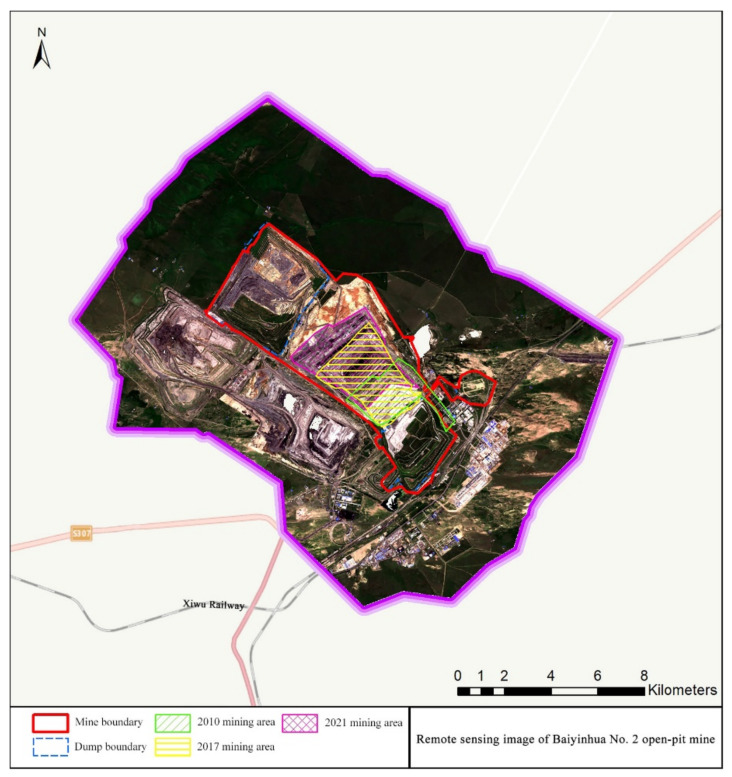
Remote sensing image of Baiyinhua No. 2 Open-pit Mine.

**Figure 3 ijerph-19-07682-f003:**
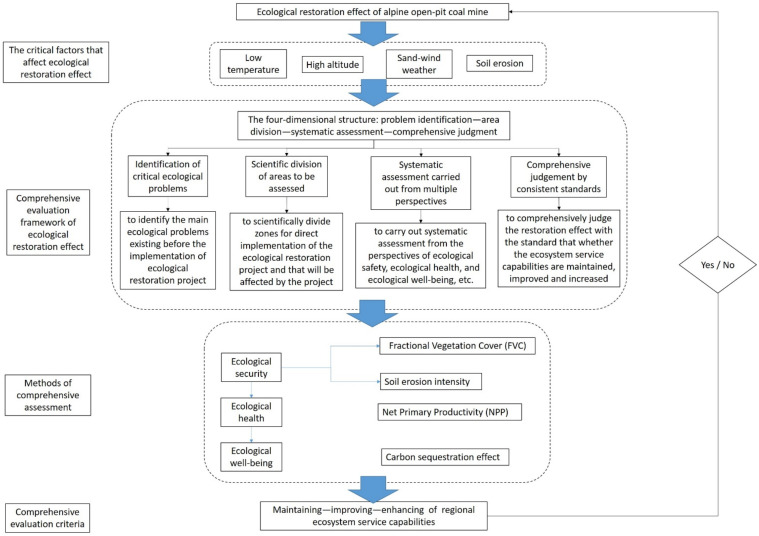
Ecological restoration effect assessment model of an alpine open-pit coal mine.

**Figure 4 ijerph-19-07682-f004:**
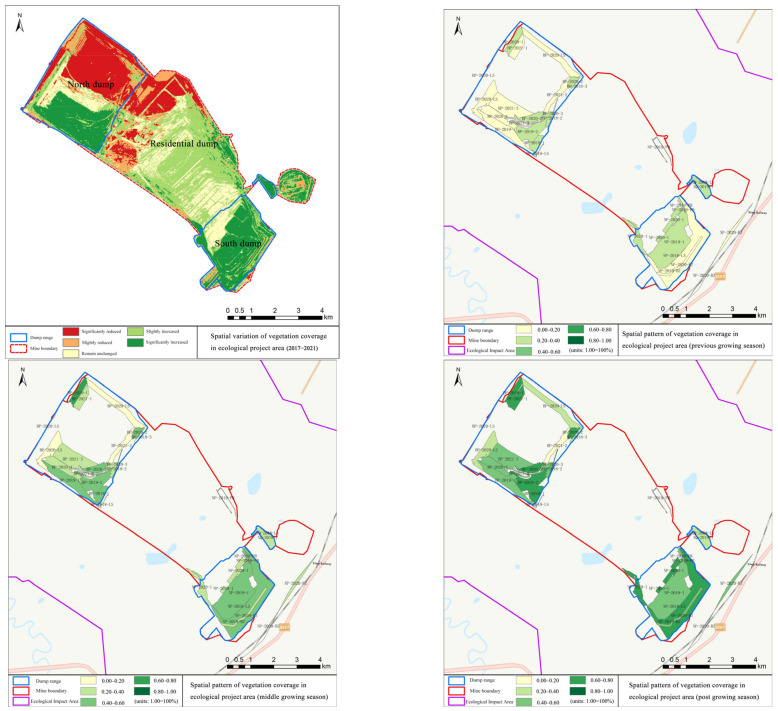
Spatial variation of vegetation coverage in the growing season in the ecological project area.

**Figure 5 ijerph-19-07682-f005:**
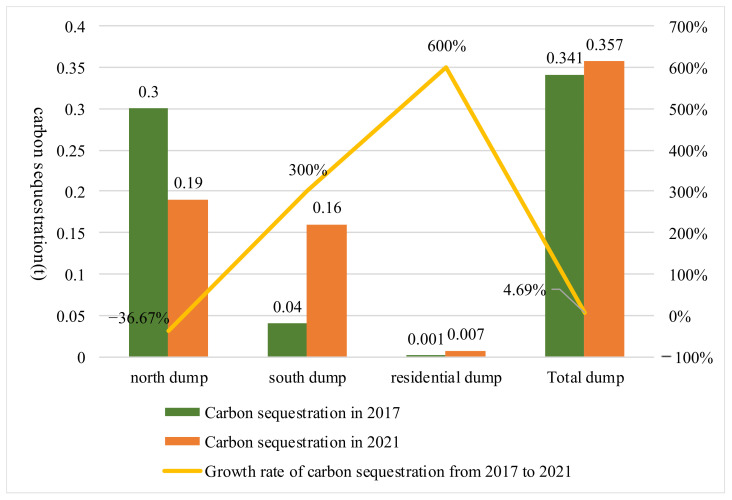
Changes in carbon sequestration in the ecological project area from 2017 to 2021.

**Figure 6 ijerph-19-07682-f006:**
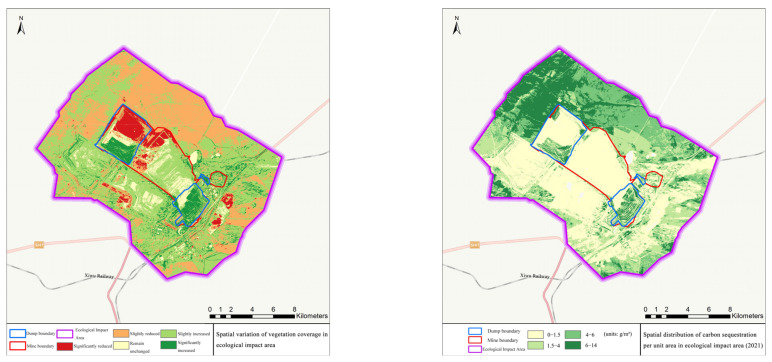
Spatial variation of vegetation coverage and changes of net primary productivity in the ecological impact area.

**Figure 7 ijerph-19-07682-f007:**
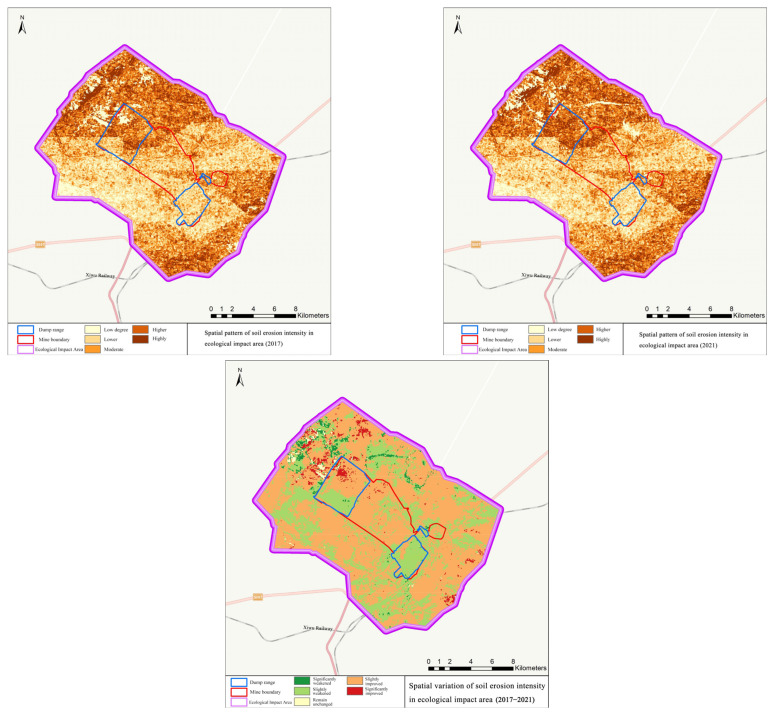
Spatial pattern of soil erosion intensity in the ecological impact area.

**Figure 8 ijerph-19-07682-f008:**
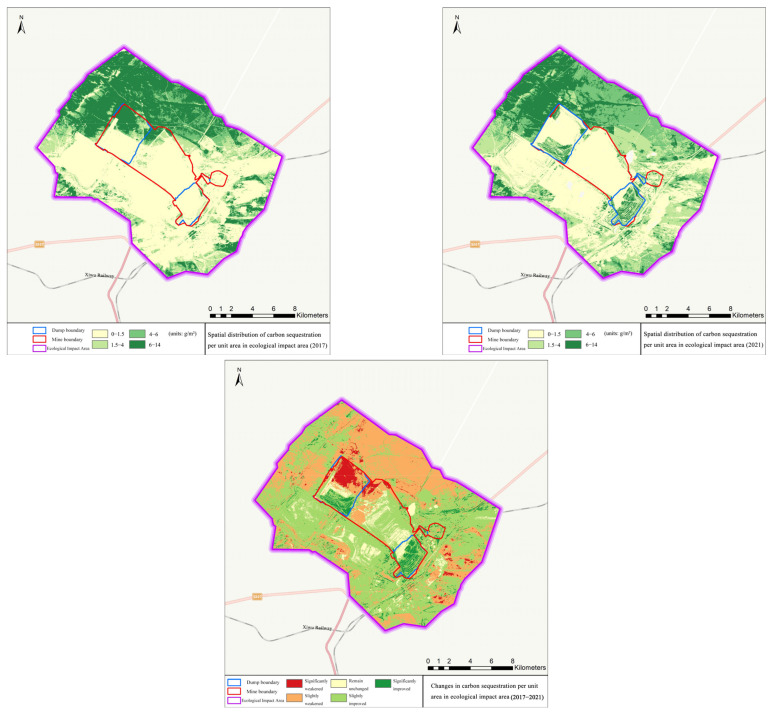
Changes of carbon sequestration effect in the ecological impact area before and after the implementation of ecological restoration projects.

## Data Availability

Not applicable.

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
