# Peer review of "Research on Ecological Effect Assessment Method of Ecological Restoration of Open-Pit Coal Mines in Alpine Regions"

_ijerph, 2022, doi:10.3390/ijerph19137682_

Round 1
Reviewer 1 Report
The revised manuscript has modified the related issues. I think it could accepted in present form.
Reviewer 2 Report
This is an interesting work to analysis the implementation effect of ecological restroation project, using Baiyinhua No.2 Open-pit Mine as a case study. The paper is well written, and the methods and results section have been written easily and understandable. Here are some suggestions for your further revision.
1. Suggest to combine Fig.1 and Fig. 2 into one figure with two sub-figure.
2. Suggest to combine Fig.3 and Fig. 4 into one figure with two sub-figure.
3. In Table 2, please adding the level of each landscape patttern index. Not all the indexes can be calculated at two levels. Besides, please adding the calculation equations for each index.
4. In line 224: please provide detail information for the data sources.
5. please reorganize the figures. figures with the same theme should be placed in the same figure, not in separate figure in different pages, such as figure 9, 10, 11,12. at least, the same figure should be in one page.
6. Please reorganize the conclusion section, extract and purify the most important conclusions.
Reviewer 3 Report
I have read the paper and my conclusion is that the paper is not suitable for publication.
Firstly, the manuscript is too long (38 pages) and contains too many figures (30 individual figures, some grouped together) many of which are superfluous (e.g. Figs 1-3 show more or less the same thing), or are difficult to interpret in isolation (e.g. Fig 5 shows spatial variation of vegetation, it shows areas reduced or unchanged but does not say relative to what baseline). All the figures require, at the very least, more information to explain what they are trying to show (e.g. Fig on page 11-13 contains codes (e.g. NP-2019-PENP) which are not explained, and the legend (repeated twice in each sub-Fig) does not give the units of vegetation coverage, we might assume 1.0 is 100% but we do not know from the Fig).
This complexity is excessive compared to the scientific content and originality of the study and, as a result, it is very hard to understand the context of the study.
More importantly, this manuscript is a report of an environmental assessment rather than scientific paper and lacks integrated outcomes.
Although this study aims to build a multi-scale and multi-dimensional comprehensive ecological effect assessment system, it is not achieved in this paper and it is not enough for a scientific manuscript to report and describe these topics separately. I do not see any scientific synthesis or progress in the manuscript.
I agree that to convert this manuscript to a publishable paper would in my opinion, require significant cuts to the length and content and very clear presentation of aims, objectives, methodology and clear scientific outcomes which should both cover the individual site but also how the methodology could be applied to other alpine situations.
Author Response
Please see the attachment.

This manuscript is a resubmission of an earlier submission. The following is a list of the peer review reports and author responses from that submission.
Round 1
Reviewer 1 Report
The open pit mines, very convenient from the point of view of economic costs, are, from the point of view of the visual impact, especially in delicate and naturalistic contexts, very devastating. This study has the merit of having determined that the ecological advantages of their restoration could be very numerous. The economic amount of any restoration programs remains to be determined, however this aspect is beyond the objectives of this paper.
1) The main problem addressed by this paper is the examination of the ecological effects deriving from the implementation of an open pit mine restoration project.
2) The topic is overall worthy of being appreciated for confirming, if scientifically necessary, the environmental benefits resulting from reforestation projects.
3) The current state of scientific research regarding this issue does not allow for a broad and detailed judgment to be expressed on the multidimensional ecological effects of any environmental restoration of opencast mines. This study fills this serious gap.
4) The conclusions and the bibliography are congruous with respect to the contents of the paper.
Reviewer 2 Report
Comprehensive ecological effect assessment of ecological restoration of open-pit coal mines in alpine regions: A case study on Baiyinhua No.2 open-pit coal mine
Authors
Meng Yuan , Jingyi Ouyang , Shuanning Zheng , Ye Tian * , Ran Sun , Rui Bao , Tao Li , Tianshu Yu , Shuang Li , Di Wu , Yongjie Liu , Changyou Xu , Yu Zhu
Manuscript ID: ijerph-1661141
General comments
This manuscript is a technical report which describes the application of six selected indicators to assess the effects of the restoration of Baiyinhua No.2 open-pit coal mine.
The work applies standard techniques, which are used worldwide in the field of mining area reclamation assessment. The selection of indicators is subjective in this type of analysis. It is difficult to argue over the selection of indicators. The lack of developed universal standards does not impose restrictive methods of the restoration assessment. Moreover, an environmental peculiarity of each mining area and long-term restoration processes determine the selection of indicators. As the authors pointed out, that the manuscript is a report of results of short-term restoration and the research is in need of continuation.
The manuscript does not provide any novelty from the methodological point of view. The results will attract a little attention from an international research audience.
Specific comments
Section 2
Lines 72-73: “…The open pit coal field is distributed from northeast 72 to southwest,…” – in fig. 1 and fig. 2 the mining field distribution is from the NW to the SE, therefore this sentence is misleading
Line 74: “… 628 hm2…, but
in Line 73: you use km2 - please decide which unit you will use,
Section 3.5
Lines 157-158: equation (4) - units are not explained anywhere, are they?
Figures
The manuscript requires a significant revision of the quality of the figures.
- in all figure there are Chinese descriptions - sorry, but Chinese words should be translated in English,
- all figures are of excessively low quality,
- explanations are illegible
- the scale in figures is in kilometres but in the text authors use hm (hectometres) – correct please.
Reviewer 3 Report
This is an interesting study to analyzing the ecological impacts of open-pit coal mines in alpine regions. However, there are still some comments on this study.
- Abstract section: please clarify not only research importance but also research gap in the first two sentences.
- Abstract section: please clarify research methods.
- Abstract section: what’s is the meaning of ‘etc.’ in line 25?
- Abstract section: what’s is the meaning of ‘etc.’ in line 29?
- Abstract section: suggest to change the last sentence.
- Introduction section: there is no information about existing relevant research or research gap. Just one sentence “In current research, comprehensive …… still insufficient.” Please provide evidence.
- Please change the base map in Figure 1 and Figure 4.
- Please add the meaning of BP in the caption of Figure 2.
- Please add the explanation of each ecosystem service in Section 3.3.
- Please explain the meaning of landscape metrics in Section 3.4.
- Suggest to move Section Discussion in the front of Conclusion.
- Please shorten the Conclusion section.
- Please add the references in line 322.
- There is lack of the discussion of research results in section 6.
Reviewer 4 Report
General comment:
The comprehensive assessment of ecological restoration in alpine regions and mines is an important issue. This paper analyzed the implementation effect of the ecological restoration project of Baiyinhua No.2 Open-pit Mine through vegetation coverage, net primary productivity, ecosystem services, landscape pattern index, soil erosion, and carbon sequestration effect. It is a very promising and potentially valuable research. Unfortunately, although the author emphasizes “multi-dimensional” “comprehensive” assessment, the actual assessment methods and process are only the analysis of single factor assessment, and do not reflect the content of “comprehensive” assessment. Moreover, there are a lot of spelling mistakes and formatting irregularities in the paper.
Introduction:
This section lacks a systematic review of research progress. There is no comprehensive summary of relevant scientific research methods and classical findings. The scientific problems are not clear.
Overview of the study area and data sources:
Description of methods used is not scientific, please re-write and give necessary references.
Methods used must be explained in detail way, allowing re-analysis, all references given to the indices used.
Line 92-98: Please provide references for the original data sets that you used. Descriptions of major data sources are too brief and cannot comprehensively support the data needs of relevant assessment indicators. Moreover, it should provide the url and access time of Geospatial Data Cloud website.
Line 115-117: what is the basis of “comprehensive” assessment system establish? Why author selected these six indicators but not others?
Line 120: It is a mistake format of “Li miaomiao et al.”, the correct format should be “Li et al.”, similar mistakes are very common in the paper, please check carefully for corrections. Besides, “Li miaomiao et al” is just only one of the user and references of the pixel dichotomy model, not its developer. Please refer to relevant literature. Scientific citation of normative references in the paper.
Line 131: This formula seems to be missing some operators, please check it.
Line 156: More information must be given about equations or methods of each factor. how these factors (R, K, LS, C, P) were obtained. For example, the rainfall erosivity factor, How many rain stations did the author use? Daily rainfall? Monthly rainfall? Average annual rainfall? What method was used to calculate “R”?
Line 161-168: How do the authors determine the relevant parameters and indicators when calculating carbon sequestration based on InVEST model? And the sources and references of relevant parameters?
Research results:
The resolution of Figure 5,6,8,9,10,11,12,13,15 is too low, and the images are particularly fuzzy, so it can hardly see any effective labeling information.
Conclusions:
Conclusion is too long, It should be reduced by min. 50%-80%. now it is condensed results+discussion.
Back matter
Author contributions, funding, institutional review board statement... are missing, these should be completed according to Template requirements.
Please use Template and format references as required by the journal style
Clear mistakes, e.g., Line 73: 510km2 (‘2’ should be the superscript) , Line 74, Line 75, etc. check throughout the text.
Referencing in the text is not according Template, please use long dash. Line 79, use [8–9] instead of [8-9], check throughout the text.